# Prediction of the Molecular Subtype of IDH Mutation Combined with MGMT Promoter Methylation in Gliomas via Radiomics Based on Preoperative MRI

**DOI:** 10.3390/cancers15051440

**Published:** 2023-02-24

**Authors:** Yongjian Sha, Qianqian Yan, Yan Tan, Xiaochun Wang, Hui Zhang, Guoqiang Yang

**Affiliations:** 1Department of Radiology, First Hospital of Shanxi Medical University, Taiyuan 030001, China; 2Xi’an No.3 Hospital, Affiliated Hospital of Northwest University, Xi’an 710018, China

**Keywords:** glioma, isocitrate dehydrogenase mutation (IDH mut), O^6^-methylguanine-DNA methyltransferase promoter methylation (MGMT meth), magnetic resonance imaging (MRI), radiomics

## Abstract

**Simple Summary:**

Glioma is the most common primary brain tumour and main cause of death of those with primary brain tumours. Multiple studies have demonstrated that glioma patients with the molecular subtype of isocitrate dehydrogenase mutation (IDH mut) combined with O^6^-methylguanine-DNA methyltransferase promoter methylation (MGMT meth) have good overall survival and/or progression-free survival and can benefit from temozolomide (TMZ) chemotherapy. However, this predictor is obtained through invasive pathologic methods. Radiomics can noninvasively excavate high-throughput features based on preoperative MRI images, and a radiomics marker mapping to a tumour molecular marker can be constructed through reasonable algorithms. The objective of this study was to establish a model for predicting the molecular subtype of isocitrate dehydrogenase mutation combined with O^6^-methylguanine-DNA methyltransferase promoter methylation in gliomas using a noninvasive radiomics model based on preoperative MRI. The model can effectively provide an important auxiliary value for the accurate diagnosis of tumour molecular typing, decision-making of the chemotherapy drug temozolomide and prognosis assessment in clinical management.

**Abstract:**

Background: The molecular subtype of IDH mut combined with MGMT meth in gliomas suggests a good prognosis and potential benefit from TMZ chemotherapy. The aim of this study was to establish a radiomics model to predict this molecular subtype. Method: The preoperative MR images and genetic data of 498 patients with gliomas were retrospectively collected from our institution and the TCGA/TCIA dataset. A total of 1702 radiomics features were extracted from the tumour region of interest (ROI) of CE-T1 and T2-FLAIR MR images. Least absolute shrinkage and selection operator (LASSO) and logistic regression were used for feature selection and model building. Receiver operating characteristic (ROC) curves and calibration curves were used to evaluate the predictive performance of the model. Results: Regarding clinical variables, age and tumour grade were significantly different between the two molecular subtypes in the training, test and independent validation cohorts (*p* < 0.05). The areas under the curve (AUCs) of the radiomics model based on 16 selected features in the SMOTE training cohort, un-SMOTE training cohort, test set and independent TCGA/TCIA validation cohort were 0.936, 0.932, 0.916 and 0.866, respectively, and the corresponding F1-scores were 0.860, 0.797, 0.880 and 0.802. The AUC of the independent validation cohort increased to 0.930 for the combined model when integrating the clinical risk factors and radiomics signature. Conclusions: radiomics based on preoperative MRI can effectively predict the molecular subtype of IDH mut combined with MGMT meth.

## 1. Introduction

Gliomas, a class of tumours that originate from glial stem or progenitor cells, are the leading cause of primary brain tumour death [1]. The five-year survival of WHO grade 2 and 3 gliomas is approximately 30–50%, and the five-year survival of WHO grade 4 gliomas is only approximately 5.6% [2,3]. In 2016, the WHO first proposed molecular classification and diagnostic criteria with isocitrate dehydrogenase (IDH) as the core in the central nervous system, which provided effective support for the accurate classification, treatment formulation and prognosis assessment of brain tumours [4,5]. IDH mutation is a positive prognostic factor, due to the overall epigenetic silencing of the glycolytic pathway that causes IDH mutant (IDH mut) gliomas to grow more slowly than IDH wild type (IDH wt) gliomas; therefore, IDH mut type predicts better overall survival (OS) and/or progression-free survival (PFS) than IDH wt [6,7]. IDH is the most critical prognostic marker, and the prognostic value of many other molecular markers (such as 1p/19q codeletion, TERT promoter mutation, ATRX loss and so on) depends on IDH [7]. O^6^-methylguanine-DNA methyltransferase (MGMT) is a DNA mismatch repair protein. The MGMT promoter methylation (MGMT meth) of the tumour silenced its expression, making it impossible to repair the damage caused by alkylating agents, so the MGMT meth suggests that patients can benefit from temozolomide (TMZ) treatment [8,9,10,11,12]. Multiple previous studies have confirmed that the subtype of IHD mut combined with MGMT meth has a longer PFS and/or OS than all of the other subtypes of IDH combined with MGMT [13,14,15]; patients with this subtype also benefit more from TMZ treatment [16,17,18]. Therefore, the accurate diagnosis of the subtype of IHD mut combined with MGMT meth in glioma is of great value for the TMZ chemotherapy treatment decision and prognosis evaluation of gliomas.

At present, the gold standard for obtaining tumour biomolecular markers is still via invasive pathological biopsy or surgical resection. Due to the lack of a macroscopic and holistic understanding of the tumour in preoperative biopsy, as well as the spatial heterogeneity in terms of the uneven distribution of tumour cell subsets in the tumour owing to the genetic diversity of the tumour, there is some uncertainty in obtaining samples containing actual tumour tissues [19]. At the same time, this method also has some problems, such as diagnostic lag and high price. Radiomics is a new approach based on computer image feature extraction and machine learning. It uses conventional MRI and can noninvasively obtain multidimensional and multiparameter image features of the static structure and dynamic function of tumours buried deep in tissues. Through big data mining, radiomics excavates massive high-dimensional image features such as intensity, texture and filtering features, to acquire radiomics markers that can map the biomolecular markers in tumours [20,21,22,23]. In recent years, a number of studies showed that MRI-based radiomics can accurately predict the IDH or MGMT status of gliomas with good performance [24,25,26,27,28,29,30]. To the best of our knowledge, there have been few studies using radiomics for the prediction of the molecular subtype of IDH combined with MGMT meth through the inclusion of clinical variables and multicentre validation.

Therefore, this study constructed and validated a radiomics model for predicting the subtype of IDH mut combined with MGMT meth in glioma via noninvasive radiomics based on preoperative MRI. Ultimately, the model showed important clinical value for the accurate diagnosis of molecular subtypes, decision-making regarding TMZ and assessment of prognosis for glioma patients.

## 2. Materials and Methods

### 2.1. Patients

This study retrospectively collected the data of 498 patients with glioma from our institutions, including the First Hospital of Shanxi Medical University (FHSXMU) and Shanxi Provincial People’s Hospital (SPPH), and The Cancer Genome Atlas (TCGA)/The Cancer Imaging Archive (TCIA) public databases. The two hospitals (FHSXMU and SPPH) are affiliated with Shanxi Medical University, and this clinical research was approved by the ethics committee of Shanxi Medical University (approval number: 2019LL101); informed consent was signed by all of the study participants. From October 2011 to July 2021, 293 patients were collected at the two hospitals. The TCGA/TCIA data (TCGA-LGG and TCGA-GBM datasets), which form a fully public dataset, were approved by Washington University medical ethics committee in St. Louis by the National Cancer Institute (grant no. 201108194), and we collected 205 patients from March 2011 to June 2015. The inclusion criteria of this study were as follows: (i). the patients had histopathologically confirmed glioma; (ii). preoperative MR images included postcontrast enhanced T1-weighted (CE-T1) and T2-weighted fluid inversion recovery (T2-FLAIR) sequences, with complete sequences and distinct images; and (iii). the patients had complete IDH and MGMT genes and clinical information. Ultimately, 417 patients met the inclusion criteria, including 228 in the FHSXMU and SPPH datasets and 189 in the TCGA/TCIA dataset.

According to the IDH mutation status and MGMT promoter methylation status, the glioma patients in the FHSXMU and SPPH datasets were divided into the subtype of IDH mut combined with MGMT meth (IDH mut and MGMT meth, 85 cases) and the other subtype (143 cases). The other subtype included IDH mut combined with MGMT promoter unmethylation (IDH mut and MGMT unmeth, 20 cases), IDH wild type combined with MGMT meth (IDH wt and MGMT meth, 55 cases) and IDH wild type combined with MGMT promoter unmethylation (IDH wt and MGMT unmeth, 68 cases). The FHSXMU and SPPH datasets were randomly divided into the training cohort and test cohort at a ratio of 7:3. The TCGA/TCIA dataset was used as an independent external validation cohort (IDH mut and MGMT meth, 90 cases; IDH mut and MGMT unmeth, 6 cases; IDH wt and MGMT meth, 43 cases and IDH wt and MGMT unmeth, 50 cases). Then, after tumour segmentation, radiomics feature extraction and feature dimension reduction, the radiomics model was finally established (radiomics pipeline: Figure 1).

### 2.2. Equipment

MR imaging was performed on the FHSXMU dataset using a 3.0 T scanner with an 8-channel array coil (Signa HDxt, GE Healthcare, Waukesha, WI, USA), and SPPH data were obtained using a 3.0 T scanner with an 8-channel array coil (Discovery 750, GE Healthcare, Waukesha, WI, USA). They used a uniform acquisition protocol: CE-T1 (TR/TE: 195 ms/4.76 ms; FOV: 240 × 240; layer thickness/spacing: 5.0 mm/1.5 mm; matrix: 256 × 256) and T2-FLAIR (TR/TE: 8000 ms/95 ms; FOV: 240 × 240; layer thickness/spacing: 5.0 mm/1.5 mm; matrix: 256 × 256). The CE-T1 sequence was obtained via injection of 0.1 mmol/kg gadolinium chelating contrast agent (Omniscan, GE Healthcare, Carrigtwohill, Co. Cork, Ireland).

### 2.3. IDH Genotyping and MGMT Methylation Testing

Glioma IDH mutation status and MGMT promoter methylation status in the TCGA/TCIA dataset were obtained from the TCGA and cBioPortal For Cancer Genomics “http://www.cbioportal.org/study.do?cancer_study_id=lgggbm_tcga_pub (accessed on 15 December 2019)” public databases. For glioma IDH mutation status in our institutional dataset, glioma DNA was extracted using the Simlex OUP^®^FFPE DNA Extraction Kit (TIB, Shanghai, China). PCR amplification was performed via ABI 9700 Life Technology (Thermo Fisher Scientific, Walthma, MA, USA). Finally, the results were obtained by Sanger sequencing via ABI 3500 Life Technology (Thermo Fisher Scientific, Walthma, MA, USA). For the methylation status of the MGMT promoter of glioma in our facility, we used the BisulFlash™ DNA Modification Kit (Epigentek, Farmingdale, New York, NY, USA) to modify the extracted glioma DNA with hydrogen sulphite and used a DRR006 kit (Takara, Kusatsu, Shiga, Japan) for PCR amplification. Finally, PyroMark Q96 (Qiagen, Hilden, Germany) was utilised to analyse and evaluate the methylation status of MGMT promoters in gliomas via bisulphite sequencing. The acquisition process of the above two molecular genotypes was described in detail in our previous article [31].

### 2.4. Tumour ROI Segmentation

First, T2-FLAIR images were registered with their corresponding CE-T1 images via FSL software “http://fsl.fmrib.ox.ac.uk/fsl/fslwiki/FSL (accessed on 23 September 2020)”. Next, with ITK-SNAP software “http://www.itksnap.org (accessed on 23 September 2020)”, only the tumour region was delineated on the CE-T1 image as the region of interest (ROI), and the oedema area around the tumour was excluded. Then, the outlined ROI was registered to the T2-FLAIR image (Figure 2). The delineation of the ROI was manually performed in a double-blind manner by two radiologists with more than 10 years of experience and was finally reviewed by a radiologist with more than 20 years of experience.

### 2.5. MRI Radiomics Feature Extraction

First, after resampling and intensity normalisation of MR images, 1702 radiomic features were extracted from the tumour ROI on CE-T1 and T2-FLAIR MR sequences. Each MR sequence contained 851 features, including 162 first-order features, 14 shape features and 675 texture features. Among them, texture features were composed of the following five matrices: grey-level co-occurrence matrix (GLCM), grey-level dependence matrix (GLDM), grey-level run length matrix (GLRLM), grey-level size zone matrix (GLSZM) and neighbourhood grey-tone dependency matrix (NGTDM), and their feature numbers were 126, 144, 144 and 45, respectively. The open software FAE ”http://github.com/salan668/FAE (accessed on 20 September 2020)” based on the PyRadiomics package “https://github.com/Radiomics/pyradiomics (accessed on 20 September 2020)” was used to extract radiomic features. To assess the repeatability of feature extraction, we randomly selected 60 patients for a double-blind comparison of manual segmentation by two radiologists. The intraclass correlation coefficient (ICC) was used to measure intraobserver consistency for feature extraction, with ICCs greater than 0.8 considered to indicate good consistency.

### 2.6. Establishment and Validation of the Radiomic Model

First, the radiomic features were preprocessed as follows: (i) To ensure the stability and robustness of the radiomic features, the features with ICCs lower than 0.8 were deleted. (ii) Due to the uneven composition ratio of the two molecular subtypes in the training cohort, the synthetic minority oversampling technique (SMOTE) was used to balance the data. (iii) The radiomic features were normalised via Z score transformation.

Then, the radiomic model was established as follows. First, a univariate logistic regression (LR) was used to screen significant radiomic features with *p* values < 0.05. Then, the least absolute shrinkage and selection operator (LASSO) regression algorithm was used to select the optimal feature subset, which was a set of non-zero coefficient radiomics features corresponding to the optimal adjustment parameter λ selected by 10-fold cross-validation. Ultimately, the radiomic model of the optimal feature subset was established via multivariate logistic regression, and the radiomic score (radscore) of each patient was calculated according to the model. The model performance and goodness of fit were tested via receiver operating characteristic (ROC) curves, calibration curves and the Hosmer–Lemeshow test for the SMOTE training cohort, the un-SMOTE training cohort (which was the original training cohort), the test cohort and the independent TCGA/TCIA validation cohort.

### 2.7. Establishment and Validation of the Clinical Model

The clinical model was constructed via multivariate logistic regression with a two-way stepwise regression method. The minimum Akaike information criterion (AIC) was used as the model selection criterion, and the ROC curve was used to test its performance.

### 2.8. Establishment and Validation of the Radiomic–Clinical Model and Nomogram

To further improve the performance of the radiomic model or complement it to some extent, the radscore of the radiomic model was calculated and integrated with the clinical model. Multivariate logistic regression was used to construct the radiomic–clinical combination model, and a nomogram based on this model was constructed. The performance of the combination model was tested using ROC curves. The integrated discrimination improvement (IDI) index was used to compare the differences between the radiomic model, the clinical model and the radiomic–clinical combination model and to quantify their differences.

### 2.9. Statistical Analysis

For the clinical characteristics of patients with different subtypes in the un-SMOTE training cohort, test cohort and independent TCGA/TCIA validation cohort, the continuous variables were presented as the median (second third quantile) and were compared between groups using the Mann–Whitney U test. The chi-square test was used for dichotomous or multidichotomous variables. Two-tailed *p* < 0.05 of these test methods was statistically significant. The above process was performed using SPSS software version 23.0 “http://www.ibm.com/cn-zh/spss (accessed on 19 October 2021)”.

The construction and validation of the radiomic model, clinical model and radiomic–clinical combination model were implemented using R software 4.1.2 “www.R-project.org (accessed on 22 November 2021)”. The R software packages used in this study included “UBL”, “GLmnet”, “RMS”, “VIM”, “ROCR”, “forestPlot”, “ResourceSelection”, “RMDA” and “PredictABEL”.

## 3. Results

### 3.1. Clinical Characteristics

The clinical characteristics of patients with different molecular subtypes in the un-SMOTE training cohort, the test cohort and the TCGA/TCIA external validation cohort are shown in Table 1. In the above three cohorts, age and tumour grade were significantly different among different subtypes, while sex was not significantly different.

### 3.2. Construction and Validation of the Radiomic Model

For the radiomic features of the training cohort, if their ICC was less than 0.8, they were removed. Due to the imbalance in the composition ratio of the two molecular subtypes, SMOTE was used to balance the data. Finally, Z score transformation was performed for the remaining radiomic features. Then, using univariate logistic regression analysis, 548 and 48 radiomic features were retained in CE-T1 and T2-FLAIR sequences, respectively. To select the best radiomic features and solve the overfitting problem, the LASSO algorithm was adopted. Finally, 16 radiomics features with non-zero coefficients corresponding to a double standard error of the minimum value of λ were selected (Figure 3), and the radiomic model was established using multivariate logistic regression (Figure 4).

To ensure the robustness and reliability of the radiomic model, we used the un-SMOTE training cohort to verify it. The results showed that the AUCs of the radiomic model in the SMOTE training cohort, the un-SMOTE training cohort, the test cohort and the independent TCGA/TCIA validation cohort were 0.936, 0.932, 0.916 and 0.866, respectively (Figure 5). For the radiomic model, the Cox-Snell R^2^ and Nagelkerke R^2^ in the un-SMOTE training cohort were 0.495 and 0.676, respectively. The relative evaluation results of the radiomic model in the SMOTE training cohort, the un-SMOTE training cohort, the test cohort and the independent TCGA/TCIA validation cohort are shown in Table 2.

The calibration curve was used for the visualisation of the radiomic model on the SMOTE training cohort, the un-SMOTE training cohort, the test cohort and the independent TCGA/TCIA validation cohort. Meanwhile, the Hosmer—Lemeshow test was used to examine the goodness of fit of the radiomic model in different cohorts, and the *p* values were 0.555, 0.792, 0.105 and 0.427, respectively (Figure 6). The radiomic model showed good agreement across the above cohorts.

### 3.3. Construction and Validation of the Clinical Model

To ensure the independence of each clinical risk factor and the goodness of fit of the clinical model, a logistic model was established using the two-way stepwise regression method, and the minimum AIC was used as the model selection criterion to construct the clinical model. Finally, age and tumour grade were included in the clinical model (Figure 7). Additionally, the AUCs of the clinical model in the un-SMOTE training cohort, the test cohort and the independent TCGA/TCIA were calculated, and the values were 0.692, 0.695 and 0.896, respectively (Figure 8).

### 3.4. Construction and Validation of the Radiomic–Clinical Model

To further improve the model performance or complement each model to some extent, the radscore calculated by the radiomic model was integrated with the clinical model, and the radiomic–clinical combined model was constructed via multivariate logistic regression (Figure 9). Interestingly, in the un-SMOTE training cohort, only the radscore was statistically significant. In the TCGA/TCIA independent validation cohort, the radscore and clinical variables were all statistically significant. To verify the performance of the combination model, the AUCs of the un-SMOTE training cohort, the test cohort and the independent TCGA/TCIA cohort were calculated, and the values were 0.936, 0.918 and 0.930, respectively (Figure 10).

### 3.5. Comparison and Visualisation of Different Models and Construction of the Nomogram

By using the IDI index, we calculated whether there was any difference between the different models in the un-SMOTE training cohort, the test cohort and the independent TCGA/TCIA validation cohort, and quantified their differences (Table 3). In the un-SMOTE training cohort and the test cohort, the radiomic model and the radiomic–clinical combined model were superior to the clinical model, with gains of approximately 45.9% and 46.6% and 46.7% and 46.6%, respectively. Although there was no statistically significant difference between the radiomic model and the radiomic–clinical combination model, the combination model still had the highest IDI index, even though the increase was negligible. In the independent TCGA/TCIA validation cohort, there was no significant difference between the radiomic model and the clinical model (*p* = 0.063). There were significant differences between the combined model and the clinical model and between the combined model and the radiomic model, and their IDI gains were 11.8% and 20.4%, respectively. In conclusion, the radiomic–clinical model had the best effect, followed by the radiomic model and the clinical model. Different models were visualised in different cohorts through decision curves (Figure 11). Finally, the nomogram of the radiomic–clinical combined model based on the TCGA/TCIA validation cohort was established and used for the visual risk prediction of molecular subtypes (the IDH mut and MGMT meth subtype was labelled one in this study, which was at a high level). Therefore, the high-risk group identified by the nomogram corresponded to this subtype (Figure 12).

## 4. Discussion

This study explored the application of radiomics based on preoperative MRI to establish a radiomic model for predicting the molecular subtype of IDH mut combined with MGMT meth in glioma. The results showed that the model had good predictive performance. Moreover, the predictive performance of the model was improved to some extent by incorporating clinical independent risk factors.

In recent years, the application and improvement of molecular subtyping has promoted the accurate diagnosis and individualised treatment of gliomas. IDH is the most critical molecular marker for prognosis evaluation [32]. The prognostic value of many other classic molecular markers (such as 1p/19q-codeleted [33], ATRX loss [34], TERT promoter mutations [35], EGFR amplification [36] and so on) depends on the mutation status of IDH. In addition, the MGMT promoter methylation is a strong prognostic biomarker in patients with gliomas treated with TMZ, because the alkylating agent resistance of glioma cells is mainly mediated by MGMT, and epigenetic silencing of the MGMT gene by the promoter methylation compromises this DNA repair and increases chemotherapy sensitivity [37]. Recently, many studies have shown that the subtype of IHD mut combined with MGMT meth is critical for the TMZ chemotherapy decision and prognosis evaluation [12,15,16]. In addition, the serum lactate dehydrogenase (LDH) is likely to become a critical biomarker for new therapy options to block certain metabolic pathways and stop tumour growth [38,39]. Compared with the molecular testing based on tumour tissue or serum samples, radiomics based on MRI provides a noninvasive method for the preoperative prediction of molecular subtype. Radiomics has been widely used in the study of IDH or MGMT single molecular markers prediction [24,25,26,27,28,29,30]. Very little radiomics research has been conducted to predict the molecular subtype of IDH mut combined with MGMT meth to support more precise diagnosis and treatment decisions.

At present, in the radiomics study of glioma IDH mut combined with MGMT meth as a combined predictor, only Zhang et al. have performed a study using an automatic machine learning radiomics method [40]. Despite the high performance of multiple models in the study, the test cohort was not strictly independent since 4-fold cross-validation was used. Moreover, the study lacked multicentre validation and did not include clinical risk factors. In our study, the FHSXMU and SPPH datasets were combined and randomly grouped at a ratio of 7:3 to ensure the complete independence of the test cohort. To verify the reliability and generality of the radiomics model, the TCGA/TCIA dataset was used as an independent external validation cohort from a multicentre perspective. In addition, the inclusion of clinical variables improved the performance of the radiomics model to a certain extent. Jian et al. obtained the comprehensive sensitivity and specificity of IDH or MGMT status in glioma through a meta-analysis of 44 papers. In general, the sensitivity of the probability that two molecular markers are both positive is obtained by multiplying their respective sensitivities together. The probability of the specific simultaneous positivity of two molecular markers was calculated as before. Therefore, the comprehensive sensitivity and specificity probabilities of the co-occurrence of IDH mut and MGMT meth calculated in this way were both approximately 0.7 [41]. This was under the premise that different modelling methods, feature duplication or correlation could be excluded, and different models could be directly fitted. The sensitivity and specificity of the study in our three cohorts were better than this value (greater than or equal to 0.8).

To further improve the performance of the radiomics model, we constructed a radiomics model, clinical model and radiomics–clinical combined model by incorporating clinical risk factors. Moreover, we compared the differences between them and quantified the magnitude of the differences, further affirming the indisputable value of the radiomics model (that is, overall, the radiomics model outperformed the clinical model, and the combined radiomics–clinical model performed the best). At the same time, for these differences, we carried out an intuitive and visual display via IDI and decision curve analyses. In univariate statistical analyses of clinical variables in the un-SMOTE training cohort, test cohort and TCGA/TCIA cohort, age and tumour grade were statistically significant when comparing clinical variables among different subtypes (*p* < 0.05, Table 1). In addition, when multiple clinical variables were fitted to the clinical model via multivariable logistic regression, age and tumour grade were ultimately retained (*p* < 0.05, Figure 7). The above results confirm the relevance and importance of these two clinical variables with the molecular subtype of IDH mut combined with MGMT meth. Interestingly, when these two variables and the radscore calculated by the radiomic model were integrated into the radiomic–clinical combination model in the un-SMOTE training cohort, only the radscore was statistically significant via multivariate logistic regression. Meanwhile, age, tumour grade and radscore were all significantly different via multivariate logistic regression in the TCGA/TCIA validation cohort. However, several previous studies have confirmed that age and tumour grade are independent risk factors for glioma and have a positive impact on survival [42,43,44]. Therefore, we still included these two clinical variables in the radiomics–clinical combined model and achieved good results in the TCGA/TCIA cohort (the AUC increased from 0.866 in the radiomics model to 0.930 in the combined model). In addition, the sufficient sample size (189 cases) of the TCGA/TCIA dataset further confirmed the value of these two clinical variables.

To solve the problem of an unbalanced sample composition ratio, we introduced the SMOTE algorithm to balance the data. Its advantages are as follows: compared with the random selection of original data via oversampling, this algorithm interpolates any single sample in a few samples with K adjacent samples (usually five by default) to improve the generalisation ability of the classifier. At the same time, compared with only undersampling of most classes, this algorithm can achieve a better classifier performance [45,46]. Of course, SMOTE also has drawbacks, such as the noise and sample replication problem caused by the choice of the K-nearest neighbour [47]. Whether the sample information can truly reflect the population is the key to the quality of the model. The SMOTE algorithm solved the sample imbalance problem to some extent, which was also reflected in the good performance of the radiomics model in our study (the AUC of the SMOTE training cohort was 0.936). In addition, to ensure the robustness of the model built on the training cohort after SMOTE, we used the training cohort without SMOTE (i.e., the original training set) for verification, and the results further confirmed the excellent performance of the model (the AUC of the training cohort without SMOTE was 0.932).

There were also some limitations in this study. (a). There were relatively many features included in the radiomics model. (b). There was an insufficient collection of clinically relevant variables. Therefore, in the construction of the radiomics model, as few significant radiomics features as possible that could explain the dependent variable should be selected by using several algorithms and ensuring that each step of feature dimension reduction is statistically reasonable. However, it should also be considered that the combination of too many dimensionality reduction algorithms may lead to insufficient model fitting due to too-strict and harsh feature screening. At the same time, for the construction of a clinical model, more clinically relevant variables should be collected to improve the performance of the clinical model. In the construction of the final radiomics–clinical combined model, the radiomics model and the clinical model were complementary to each other to further improve the predictive performance of the combined model.

## 5. Conclusions

In conclusion, this study constructed a radiomics model for predicting the molecular subtype of IDH mut combined with MGMT meth in glioma via radiomics based on preoperative MRI. The results showed that the model had good predictive performance. The radiomic model integrating clinical independent risk factors improved the predictive performance of the model to a certain extent. It may provide important auxiliary value for the accurate preoperative diagnosis of molecular subtypes, TMZ chemotherapy decisions and the evaluation of OS and/or PFS in patients with glioma.

## Figures and Tables

**Figure 1 cancers-15-01440-f001:**
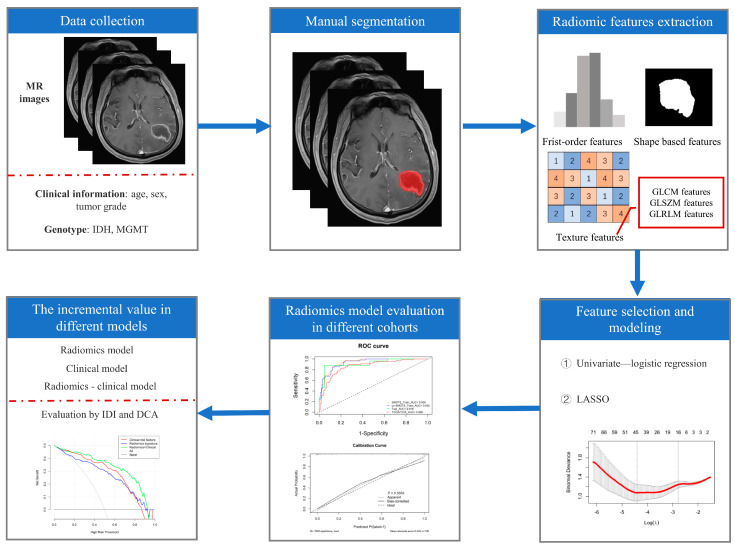
Radiomics pipeline. Note: LASSO (least absolute shrinkage and selection operator), IDI (integrated discrimination improvement index) and DCA (decision curve analysis).

**Figure 2 cancers-15-01440-f002:**
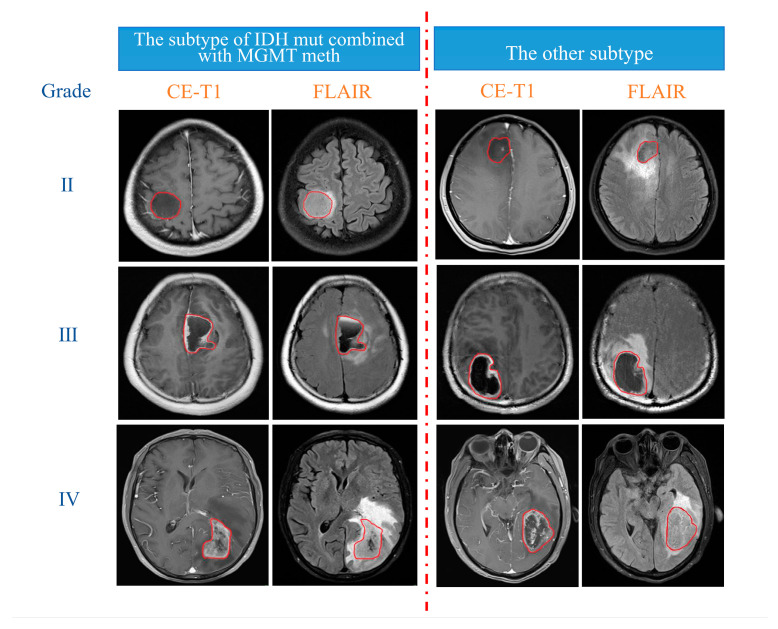
Delineation details of the ROI in irregular red circles.

**Figure 3 cancers-15-01440-f003:**
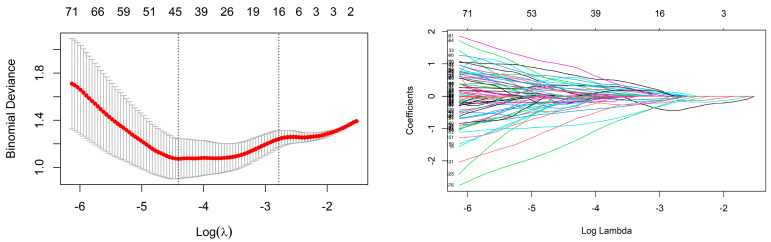
LASSO features selection.

**Figure 4 cancers-15-01440-f004:**
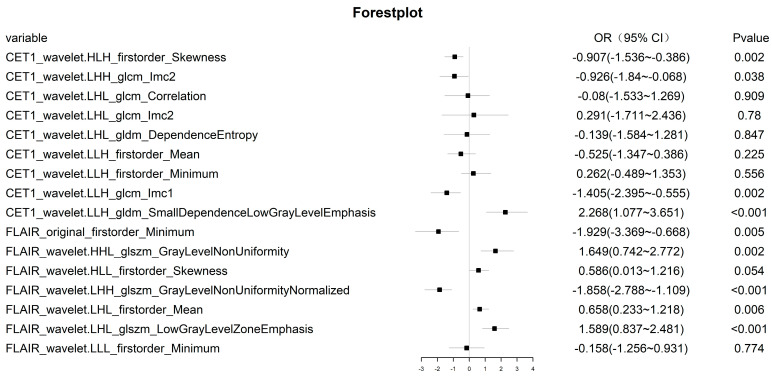
Sixteen features of the radiomic model, and the OR values of each radiomic feature and its 95% CI and *p* value.

**Figure 5 cancers-15-01440-f005:**
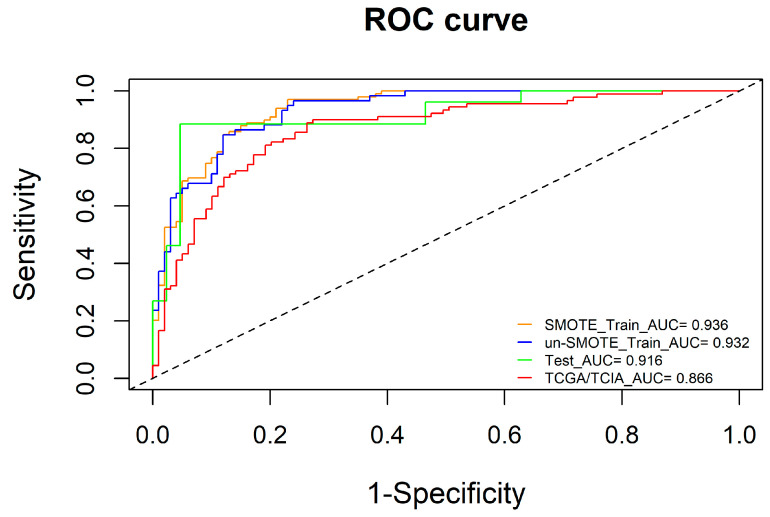
ROC curves of the SMOTE training cohort, un-SMOTE cohort, test cohort and independent TCGA/TCIA cohort.

**Figure 6 cancers-15-01440-f006:**
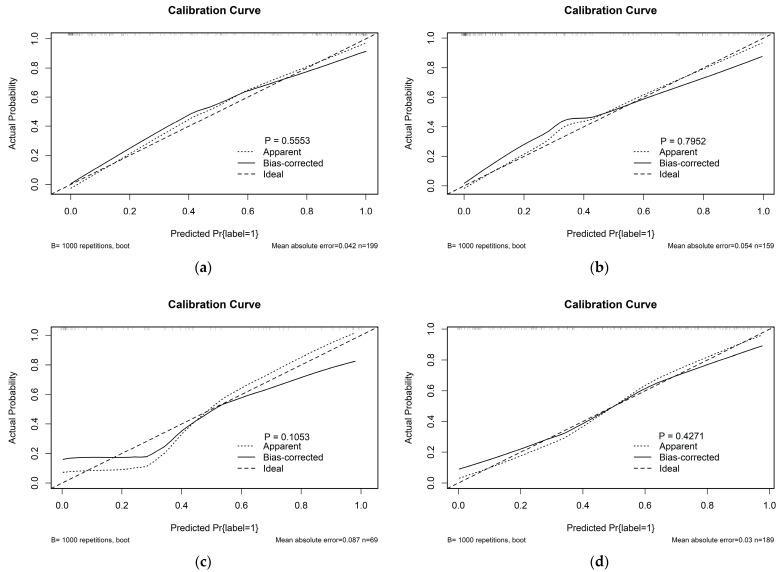
Calibration curves of the radiomic model in (**a**) the SMOTE training cohort; (**b**) the un-SMOTE training cohort; (**c**) the test cohort and (**d**) the independent TCGA/TCIA validation cohort, and their corresponding *p* values.

**Figure 7 cancers-15-01440-f007:**
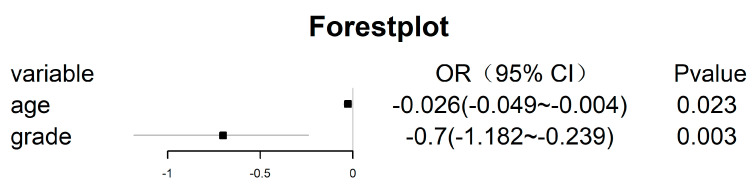
Variables included in the clinical model with the OR values of each variable and its 95% CI and *p* values.

**Figure 8 cancers-15-01440-f008:**
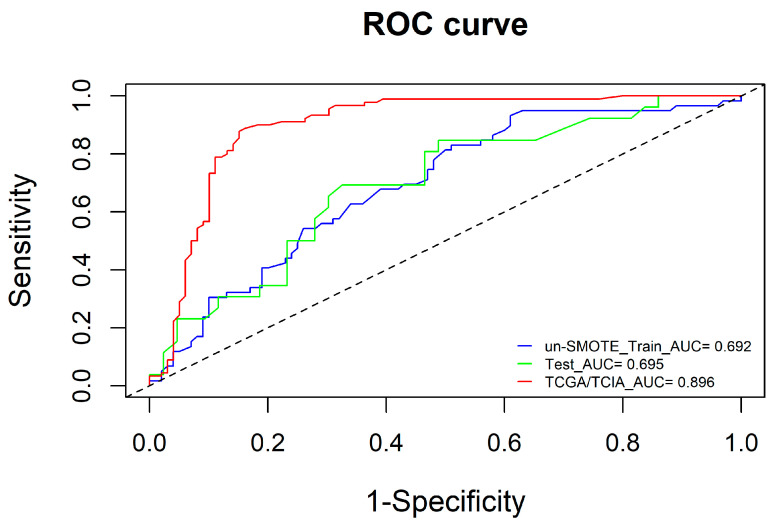
ROC curves of the clinical model in the un-SMOTE training cohort, test cohort and independent TCGA/TCIA validation cohort.

**Figure 9 cancers-15-01440-f009:**
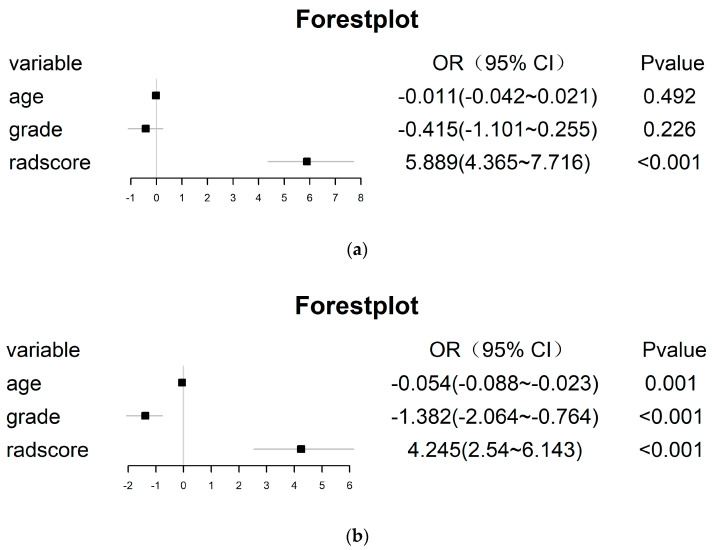
All features of the radiomic–clinical combined model, and the OR value and its 95% CI and *p* value of each feature. (**a**) Training cohort. (**b**) TCGA/TCIA independent validation cohort.

**Figure 10 cancers-15-01440-f010:**
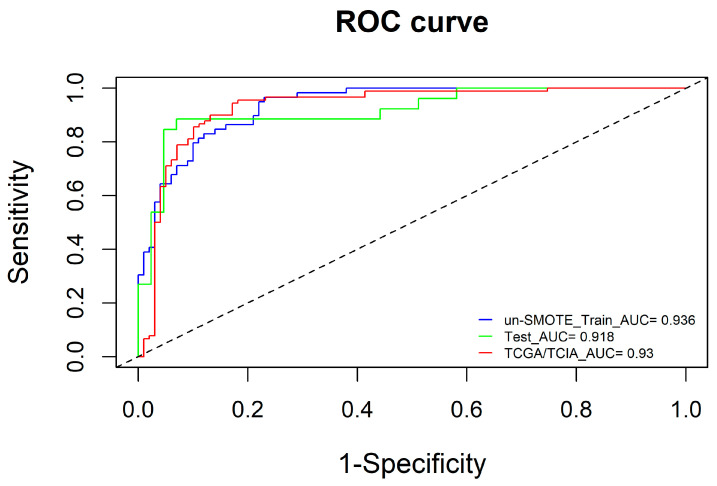
ROC curves of the combined model in the un-SMOTE training cohort, test cohort and independent TCGA/TCIA validation cohort.

**Figure 11 cancers-15-01440-f011:**
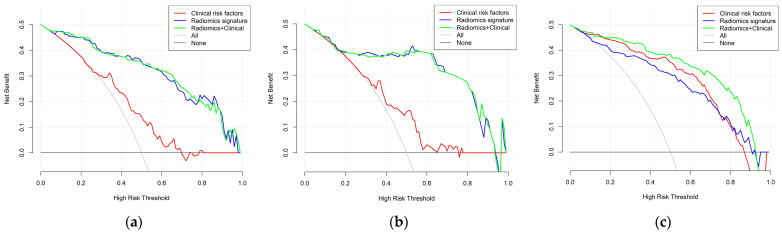
Decision curve analysis of the three models: (**a**) un-SMOTE training cohort, (**b**) test cohort and (**c**) TCGA/TCIA validation cohort.

**Figure 12 cancers-15-01440-f012:**
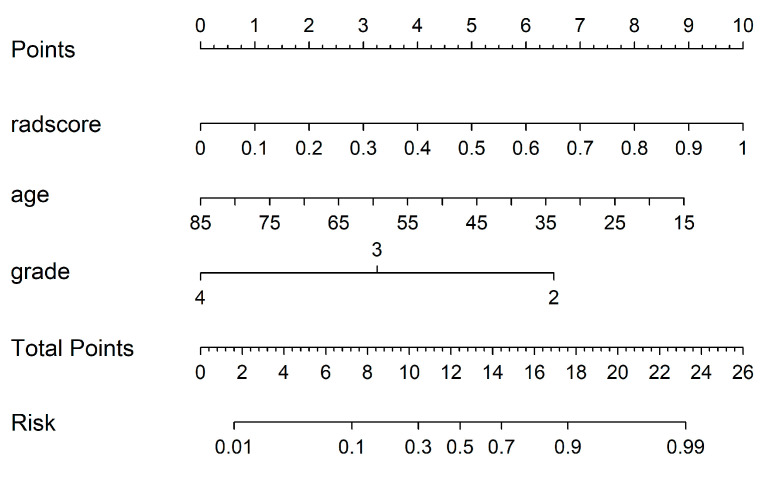
Nomogram of the radiomic–clinical model.

**Table 1 cancers-15-01440-t001:** Patient characteristics of three cohorts.

	un-SMOTE Train Cohort			Test Cohort			TCGA/TCIA Cohort		
	The Subtype of IDH Mut and MGMT Meth	The Other Subtype *	*p* Value	The Subtype of IDH Mut and MGMT Meth	The Other Subtype *	*p* Value	The Subtype of IDH Mut and MGMT Meth	The Other Subtype *	*p* Value
	(n = 59)	(n = 100)		(n = 26)	(n = 43)		(n = 90)	(n = 99)	
Age (year)	45 (37.5~52.5)	55 (43.5~62.0)	0.001	43.5 (41~56)	54 (45.5~61.5)	0.044	42.5 (33~53)	58 (50~66)	<0.001
Sex			0.707			0.555			0.934
female	23 (39%)	36 (36%)		14 (53.8%)	20 (46.5%)		46 (51.1%)	50 (50.5%)	
male	36 (61%)	64 (64%)		12 (46.2%)	23 (53.5%)		44 (48.9%)	49 (49.5%)	
Grade			0.003			0.049			<0.001
2	14 (23.7%)	12 (12.0%)		8 (30.8%)	5 (11.6%)		46 (51.1%)	7 (7.1%)	
3	29 (49.2%)	34 (34.0%)		10 (38.5%)	13 (30.2%)		38 (42.2%)	21 (21.2%)	
4	16 (27.1%)	54 (54.0%)		8 (30.8%)	25 (58.1%)		6 (6.7%)	71 (71.7%)	

* The other subtype included IDH mut combined with MGMT promoter unmethylation (IDH mut and MGMT unmeth), IDH wild type combined with MGMT meth (IDH wt and MGMT meth) and IDH wild type combined with MGMT promoter unmethylation (IDH wt and MGMT unmeth).

**Table 2 cancers-15-01440-t002:** Evaluation results of the radiomic model in the SMOTE training cohort, un-SMOTE training cohort, test cohort and independent TCGA/TCIA validation cohort.

	Accuracy	Precision	Sensitivity	Specificity	AUC	F1-Score
SMOTE training cohort	0.859	0.851	0.869	0.850	0.936	0.860
un-SMOTE training cohort	0.849	0.797	0.797	0.880	0.932	0.797
Test cohort	0.913	0.917	0.846	0.953	0.916	0.880
TCGA/TCIA validation cohort	0.968	0.793	0.811	0.808	0.886	0.802

**Table 3 cancers-15-01440-t003:** Evaluation results of IDI in the un-SMOTE training cohort, test cohort and independent TCGA/TCIA validation cohort.

Cohorts	IDI * (95% CI)	*p* Value
Training cohort		
radiomic vs. clinical	0.459 (0.368–0.551)	<0.001
radiomic–clinical vs. clinical	0.466 (0.381–0.551)	<0.001
radiomic–clinical vs. radiomic	0.007 (−0.007–0.021)	0.335
Test cohort		
radiomic vs. clinical	0.467 (0.227–0.452)	<0.001
radiomic–clinical vs. clinical	0.466 (0.343–0.590)	<0.001
radiomic–clinical vs. radiomic	4 × 10^−4^ (−0.009–0.010)	0.943
TCGA/TCIA validation cohort		
radiomic vs. clinical	−0.080 (−0.164–0.004)	0.063
radiomic–clinical vs. clinical	0.104 (0.063–0.146)	<0.001
radiomic–clinical vs. radiomic	0.184 (0.132–0.236)	<0.001

* IDI: integrated discrimination improvement

## Data Availability

The data used for the study are available upon request. The TCGA contains clinical and genetic information on glioma patients at “https://www.cancer.gov/about-nci/organization/ccg/research/structural-genomics/tcga (accessed on 15 December 2019)”; TCIA for patients with TCGA corresponding image data, address: “https://www.cancerimagingarchive.net (accessed 15 December 2019)”.

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
