# Peer review of "Prediction of the Molecular Subtype of IDH Mutation Combined with MGMT Promoter Methylation in Gliomas via Radiomics Based on Preoperative MRI"

_cancers, 2023, doi:10.3390/cancers15051440_

Round 1
Reviewer 1 Report
This is a good study however
1) the authors have made this study to be super niched to show originality thus reducing its significance.
2) lacks the most recent citations that may support and improve the introduction of the paper.
Reviewer 2 Report
Dear Authors,
Congratulations on the great work.
1. Methods are clearly defined and discussed.
2. Citations have been properly used to make a point in the current study.
3. Sample size of the study is great. This provides power to the study.
4. Methods were followed to generate significant data for gliomas detection using radiomics model.
5. The model used could effectively provide important auxiliary value for the accurate diagnosis of tumor molecular typing, decision-making of the chemotherapy drug temozolomide, and prognosis assessment in clinical management.
Please check the paper for minor spell checks. Thank you.
Reviewer 3 Report
The work is potentially interesting
However, corrections are needed to make the paper interesting and suitable for publication
1. the abstract is too broad and should be focused on the basics and shortened
2. the introduction of the work is also too broad and dysfunctional, inappropriate. The authors write about various things that are not the main topic of the work, but about IDH, they write very little and that is the main topic of the work
3. In the introductory part, the role of isocitrate dehydrogenase (IGH) and its importance in glioblastoma is not mentioned at all, and it is not stated why this is the goal and what is the advantage compared to other markers that are currently used.
4. It is necessary to add to the work permission for the use of patient data and the use of data from some database and which one???
5. A decision of the research ethics committee is required and studies are conducted on the material
6. the discussion does not include all important references, as well as the use of classic biomarkers for tumor assessment
7. Serum LDH values have always been useful as a tumor marker and the works should be cited: PMID: 26530363, PMID 35838532
8. Little has been written about methylation
9. In the discussion section, methylation is not associated with the obtained findings, which is the title of the paper, and the association with clinical data should be added, as described for other tumors in the paper PMID: 36378420
Round 2
Reviewer 1 Report
The authors have addressed my comments and have edited the manuscript accordingly.
Reviewer 3 Report
the work has been corrected